



# Particulate macronutrient exports from tropical African montane catchments point to the impoverishment of agricultural soils

Jaqueline Stenfert Kroese[1,2], John N. Quinton[1], Suzanne R. Jacobs[4], Lutz Breuer[3,4] and Mariana C. Rufino[1,2]

[1]Lancaster Environment Centre, Lancaster University, Lancaster, LA1 4YQ, United Kingdom
[2]Centre for International Forestry Research (CIFOR), c/o World Agroforestry Centre (ICRAF), Nairobi, 00100 Kenya
[3]Institute for Landscape Ecology and Resources Management (ILR), Justus Liebig University Giessen, Giessen, 35392, Germany
[4]Centre for International Development and Environmental Research (ZEU), Justus Liebig University Giessen, Giessen, 35390, Germany

*Corresponding author: Jaqueline Stenfert Kroese (J.Stenfertkroese@lancaster.ac.uk)*

**Abstract** Catchments in the tropics often generate high concentrations of suspended sediments following the conversion of forests to agriculture. The eroded fine particles are generally enriched with carbon and nutrients originating from the topsoil. Sediment-associated carbon and nutrients are an important loss to the terrestrial ecosystem and tightly connected to processes controlling riverine particulate carbon and nutrient export. Soil nutrient depletion can limit crop growth and yields, whereas an excess of nutrients in streams can cause eutrophication in freshwater systems. Streams in East Africa, with widespread land conversion are expected to receive high loads of sediment-associated carbon, nitrogen and phosphorus. In this study, we build the knowledge base for particulate carbon, nitrogen and phosphorus for contrasting land uses. Suspended sediments (time-integrated, manual-event based and automatic-event based sediment samples) were analysed for total carbon, nitrogen and phosphorus concentrations collected at the outlet of a natural montane forest, a tea-tree plantation and a smallholder agriculture catchment in western Kenya during two sampling campaigns in 2018 and 2019. Particulate carbon and nutrient concentrations were up to three-fold higher ($p < 0.05$) in the natural forest catchment compared to fertilized agricultural catchments. The higher carbon and nutrient ratios in the natural forest suggest that the particulate nutrients are of organic origin due to tighter nutrient cycles, whereas lower ratios in both agricultural catchments suggest mineral sediment sources. The findings of this study imply that with the loss of natural forest, the inherent soil fertility is progressively lost under the current low fertilization rates and soil management strategies.





# 1 Introduction

In sub-Saharan Africa, streams often have high concentrations of suspended sediments mainly due to the anthropogenic disturbance of natural ecosystems (Mogaka et al., 2006; Penny, 2009). These sediment concentrations can be particularly high in the steep highlands of East Africa, where surface soil erosion in catchments dominated by agriculture generates significantly more suspended sediments than in forested catchments (Brown et al., 1996; Stenfert Kroese et al., 2020b; Tamooh et al., 2014). The loss of organic carbon- (C) and nutrient-rich topsoil through surface runoff induced by soil erosion from agricultural surfaces (Powlson et al., 2011; Quinton et al., 2001) leads to further deterioration of tropical soils (Okalebo et al., 2005; Tully et al., 2015). Phosphorus (P) and nitrogen (N) are soil nutrients which can limit crop growth when they are in low supply, and therefore their loss from topsoils should be avoided (Pasley et al., 2019). The clays found in many of the tropical soils of East Africa contain amorphous iron (Fe), which can limit plant P availability and consequently the application of fertilizers (inorganic N and P and/ or manure) is often required to improve agricultural productivity (Mutuo et al., 1999). Soil-forming processes tend to be much slower than the rates of soil loss through erosion (Amundson et al., 2015; Evans et al., 2020), which can lead to soil and plant nutrient deficits resulting in low and stagnant crop yields (Lederer et al., 2015; Saiz et al., 2016). Soil organic matter is a critical determinant of soil fertility, providing nutrients to plants, and playing fundamental roles in soil carbon sequestration and soil water functions (Owuor et al., 2018; Weil and Brady, 2016). Processes affecting nutrient stocks and pools of soils are tightly connected to the processes controlling riverine C and nutrient fluxes.

Sediments and their associated organic C and nutrients, such as N and P (Horowitz, 2008; Johnson et al., 2018; Quinton et al., 2001) may impact streams by reducing benthic communities, primary productivity and water storage capacity of water reservoirs (Hunink and Droogers, 2011; Tamene et al., 2006) and by increasing turbidity (Stenfert Kroese et al., 2020b; Tamooh et al., 2014). In nutrient-limited freshwater systems, an excess of nutrients can cause eutrophication (Jarvie et al., 2019; Smith et al., 2017; Smith and Schindler, 2009), inducing algal blooms and promoting invasive weeds (e.g. water hyacinth) (Lung'ayia et al., 2001). When C enters the watercourse, it can be mineralised and emitted as a greenhouse gas (Marx et al., 2017b, 2017a). Nutrient-enriched sediments can turn stream waters from sink to source of nutrients through sorption and desorption processes and increased chemical and biological activity (Kreiling et al., 2019; Mainstone and Parr, 2002; Palmer-Felgate et al., 2009).

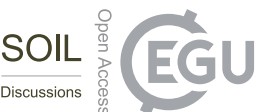

Several studies investigating the effects of land use change and agricultural intensification on the water quality

of river systems have focused on dissolved C and nutrients such as N and P (Bowes et al., 2018; Drewry et al., 2009; Smith et al., 2017). Others have shown that up to 95 % of the nutrient loads and up to 40 % of the C loads in streams are transported in particulate form, associated to suspended sediments (Moran et al., 2005; Rodríguez-Blanco et al., 2015; Scanlon et al., 2004). Sediment-associated nutrients are spatially variable (Harrington and Harrington, 2014; Withers et al., 2001) and their concentrations increase during storm events

with the rising limb of the hydrograph (Drewry et al., 2009; López-Tarazón et al., 2016). Walling et al. (1997) and Bender et al. (2018) observed that P loads mainly occur in particulate form, while N is mainly transported in dissolved form (Wang et al., 2015). Others have found that the major inputs to rivers occur as dissolved P and N loads (Harrington and Harrington, 2014). Currently, there are no studies for East Africa investigating sediment-associated carbon, nitrogen and phosphorus exports from different land uses. This is an important gap

in the knowledge because the East Africa region has a very dynamic land use system, with intensive conversion of natural ecosystems (forests, wetlands and grasslands) to subsistence and commercial agriculture (Carter et al., 2018).

This study aims to fill this knowledge gap by developing an improved understanding of the response of suspended sediments and carbon and nutrient fluxes associated with sediments under contrasting land use at

catchment scale in the headwaters of the Sondu River Basin originating in the Mau Forest Complex, Kenya. Earlier work in this area found much higher suspended sediment yields and dissolved nitrate exports in agricultural compared to forested land use (Jacobs et al., 2018; Stenfert Kroese et al., 2020b) and identified agricultural land as the main source of sediments within a smallholder agriculture catchment (Stenfert Kroese et al., 2020a). Therefore, we set out to test the hypothesis that the sediment-associated carbon and nutrient fluxes

would be higher from the more intensively managed agricultural catchments than from a natural forest catchment.

## 2 Materials and methods

### 2.1 Catchment characteristics

This study was conducted in nested catchments in the montane headwaters of the Sondu River Basin

(3,470 km$^2$) located in the South-West Mau, Western Kenya, home to one of the largest closed-canopy montane



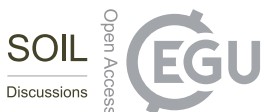

forest of East Africa (the Mau Forest Complex), and where the inherent fertility of the soils allowed the development of a tea industry in parallel to subsistence agriculture (Binge, 1962). Over the last four decades, the western Highlands of Kenya have undergone significant land use changes, whereby 25 % of the Mau Forest Complex was converted to commercial and smallholder agriculture (Brandt et al., 2018). As an important

headwater area to tributaries of Lake Victoria, the Mau Forest Complex is a critical catchment area that supplies people (approximately 5 million), livestock, wildlife and the economy with fresh water (UNEP, 2008). Lake Victoria, an important freshwater lake for five countries, has shown increased signs of eutrophication over recent years (Zhou et al., 2014), stressing the need for mitigation and control of nutrient inputs.

The study catchments are characterized by distinct land uses: (1) natural forest (NF; 35.9 km$^2$), (2) tea-tree

plantations (TTP; 33.3 km$^2$) and (3) smallholder agriculture (SHA; 27.2 km$^2$) (Figure 1 & Figure 2). The mean slope gradient of the catchments ranges between 11.6 and 15.7 %, up to a maximum of 72 % in the natural forest catchment. The streams are first- and second-order perennial streams merging to Sondu River (a sixth-order stream). The region has a bimodal rainfall pattern with a long rainy season (March-June) and a short rainy season (October-December) with a continued intermediate rainy season between the two wet seasons

(July-September). The driest months are in January and February. The mean annual rainfall is 1,979 ± 325 mm yr$^{-1}$ (period 1905-2019). Geology is composed of folded volcanics from the early Miocene. Kericho Phonolites cover the lower catchment (tea-tree plantations), followed by phonolitic nephelinites with intercalated tuffs and Mau ashes with basal tuff encompassing the natural forest, while phonolitic nephelinites comprises the upper catchment in the smallholder agriculture (Binge, 1949; Jennings, 1962). The catchments are

covered by deep (>1.8 m) and well-drained, dark-red loamy soils (Sombroek et al., 1982), characterized as mollic Andosols and humic Nitisols (ISRIC, 2004) with moderate to high amounts of organic matter (Dunne, 1979).





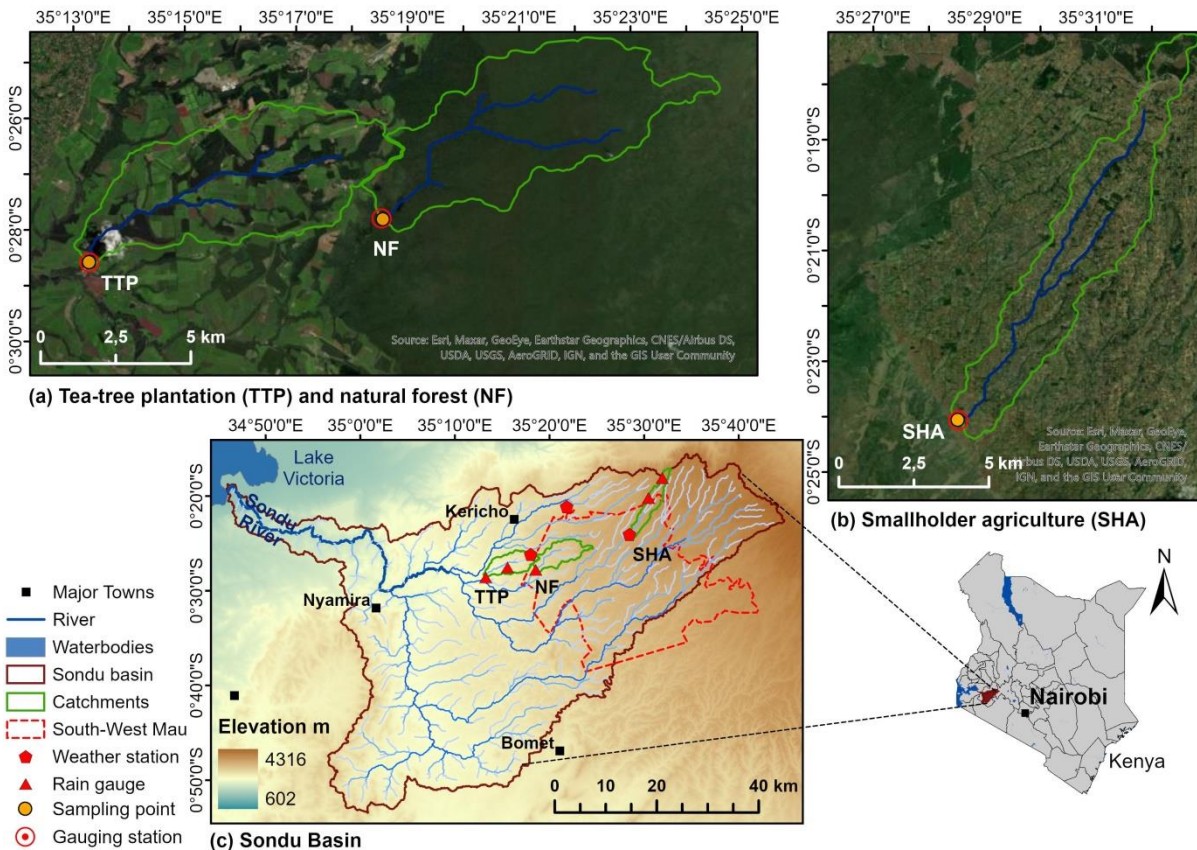

**Figure 1 (a) Tea-tree plantation (TTP) and natural forest (NF) and (b) smallholder agriculture (SHA) catchments**
**with imagery basemap (Esri, 2020) as nested catchments of the (c) Sondu River Basin with elevation (SRTM digital**
**elevation model 30 m resolution) (USGS, 2000) in the South-West Mau, Kenya.**

The natural forest catchment falls within the South-West Mau Forest part of the Mau Forest Complex. As an
afromontane mixed forest, species such as *Polyscias kikuyuensis*, *Macarangea kilimandscharica*, *Olea*
*hochstetteri*, *Casearia battiscombei* and *Fagara* spp. dominate the vegetation, transitioning to irregular patches
of bamboo forest above 2,300 m above sea level (a.s.l.) (Binge, 1962). The riparian zone transits from the forest
vegetation containing an understorey with a dense cover of shrubs and tree ferns combined with tall indigenous
tree species (Table 1).

The tea-tree plantation catchment borders the South-West Mau to the west. The catchment is characterised by
tea (*Camellia* spp.) plantations alternated with *Eucalyptus saligna* and *Cupressus lusitanica* woodlots used for
fuelwood for tea processing at the tea factories and timber production. The common practices to control soil



erosion are mulching and interplanting rows of oat grass between rows of tea during the establishment of new tea fields, cover crop establishment with mature tea trees, and terracing and sited cut-off drains. A well maintained road system along with open culverts as road drainages connects surface runoff with the riparian buffer zone. Herbicides are commonly used to control weeds. Aerial application of inorganic fertilizer is conducted two to three times per year on the tea plantations (150-250 kg N ha$^{-2}$ yr$^{-1}$ and 8-13 kg P ha$^{-2}$ yr$^{-1}$) (Jacobs et al., 2018). Riparian zones of up to 30 m from the river are commonly vegetated by native tree species densely covering the ground (Table 1).

In the upper smallholder agriculture catchment, subsistence farmers grow maize interspersed with beans, potatoes, millet, cabbage and tea (*Camellia* spp.) on farms usually smaller than one hectare. Grasslands for livestock and woodlots of *Eucalyptus saligna*, *Cupressus lusitanica* and *Pinus patula* are alternated with agricultural land. A combination of hoeing and herbicide application is used for weed control. Inorganic fertilizer is applied manually on potatoes and maize (23-45 kg N ha$^{-2}$ yr$^{-1}$ and 12-23 kg P ha$^{-2}$ yr$^{-1}$ twice a year on potatoes and once a year on maize), while manure is commonly used for cabbage and other greens. Unpaved roads, frequently used by people, livestock and motorbikes, often develop into deeply incised gullies running down slope, coupling hillslopes with the stream network. Degraded river banks with a sparse or absent riparian vegetation are prone to erosion, and in some places degraded riparian wetlands are found (Table 1).



**Table 1 Catchment characteristics under different land use natural forest, tea-tree plantations and smallholder agriculture in the South-West Mau, Kenya.**

| | | Natural forest | Tea-tree plantations | Smallholder agriculture |
|---|---|---|---|---|
| **Outlet coordinates[a]** | | 35°18'32.0472"E 0°27'47.592"S | 35°13'17.22"E 0°28'34.9176"S | 35°28'31.7316"E 0°24'4.0248"S |
| **Area (km$^2$)** | | 35.9 | 33.3 | 27.2 |
| **Elevation range (m a.s.l.)** | | 1,968-2,385 | 1,788-2,141 | 2,389-2,691 |
| **Mean slope ± SD (%)** | | 15.7±8.4 | 12.4±7.6 | 11.6±6.7 |
| **Basin order (Strahler)** | | 1, 2 | 1, 2 | 1, 2 |
| **Drainage density (km km$^{-2}$)** | | 0.48 | 0.42 | 0.64 |
| **Sediment** | **Clay %** | 81 | 76 | 87 |
| **particle size** | **Sand %** | 19 | 24 | 13 |
| **Geology** | | Phonolites | Phonolitic nephelinites | Phonolitic nephelinites and Mau ashes with basal tuff |
| **Dominant soils[b]** | | Humic Nitisols | Humic Nitisols | Mollic Andosols & humic Nitisols |
| **Vegetation** | | Afromontane mixed forest with broad-leafed evergreen trees and shrubs, grassland, bamboo | Perennial tea plantations, *Eucalyptus saligna* and *Cupressus lusitanica* woodlots | Perennial & annual crops (maize, beans, potatoes, millet, cabbage and onions, tea), woodlots, grassland |
| **Riparian vegetation** | | Forest vegetation | >30 m buffer with indigenous vegetation | Degraded riparian vegetation and wetlands, Eucalyptus woodlots |

[a] WGS 1984 UTM Zone 36S
[b] KENSOTER Geology data from the Soil and Terrain database for Kenya (KENSOTER) version 2.0

## 2.2   Continuous field monitoring

The outlet of each catchment is equipped with an automatic gauging station to measure continuously (10 minute interval) water level (m) with a radar sensor (VEGAPULS WL61, VEGA Grieshaber KG, Schiltach, Germany) and turbidity (formazin turbidity unit=FTU) using a UV/Vis spectroscopy sensor (spectro::lyser, s::can

Messtechnik GmbH, Vienna, Austria) (Figure 1). Stream discharge (m$^3$ s$^{-1}$) was obtained by using a site-specific second-order polynomial water level to discharge rating curve (Jacobs et al., 2018). Specific discharge [mm day$^{-1}$] was determined by integrating instantaneous discharge taken at 10 minute intervals over a day and relating it to the catchment area. *In situ* turbidity measurements were used to estimate suspended sediment concentrations (mg L$^{-1}$), based on a rating curve between turbidity and suspended sediment concentrations established by

Stenfert Kroese et al. (2020b). After obtaining continuous discharge and suspended sediment concentration values, suspended sediment load was determined by multiplying suspended sediment concentration and

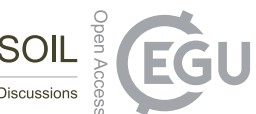

discharge for each 10 minute interval. Suspended sediment yield was calculated by integrating sediment load over time and relating it to the catchment area.

Precipitation was measured using eight automatic tipping bucket rain gauges calibrated to measure cumulative
rainfall every 10 minutes with a 0.2 mm resolution (5 tipping bucket rain gauges: Theodor Friedrichs, Schenefeld, Germany, and 3 weather stations: ECRN-100 high resolution rain gauge) (Figure 1). Thiessen polygons weighted the contribution of rainfall of every tipping bucket to each catchment. A more detailed description of sampling sites and instrumentation can be found in Jacobs et al. (2018). This study uses hydrological and sedimentological data between January 2018 and December 2019.

**2.3    Data quality assurance**

For quality assurance the turbidity, discharge and rainfall datasets were checked for anomalities. Recorded values were flagged with Not-a-Number (NaN) during malfunctioning such as (i) sensor above water level, (ii) siltation of turbidity sensors, (iii) biofilm or debris on the measurement window due to malfunctioning of automatic cleaning with compressed air, (iv) measurement gaps due to incidents of power supply failure or (v)
restricted counting of number of tips by the rain gauges by blocked funnel or spiderwebs.

Once anomalous values were replaced by NaN, local outliers were detected by the median absolute deviation (MAD) for discharge and suspended sediment:

$$MAD_i = b\, M_{i2}(|x_i - M_{i1}(x_i)|) \qquad (1)$$

where $x_i$ is the whole dataset, $M_{i1}$ is the median of the dataset and $M_{i2}$ is the median of the absolute deviation
from the dataset from its median. The standard deviation is estimated by the constant $b$ set to 1.4826 for normal distribution (Leys et al., 2013). A moving window of $k$=16 measurements around observation $x_i$ at time $t_i$ was used to detect local outliers with $x_j = (x_{i-k/2} \ldots x_{i-1}, x_{i+1} \ldots x_{i+k/2})$:

$$\frac{x_i - M_{j,i}}{MAD_{j,i}} > a \qquad (2)$$

where $a$=6 is the threshold for outlier selection, $M_{j,i}$ is the median and the $MAD_{j,i}$ is the MAD for $x_j$. There are
gaps for discharge and suspended sediment data for the smallholder agriculture catchment in September until



October 2019 due to theft of the power supply. Missing sediment data was integrated using a linear interpolation.

## 2.4    Suspended sediment sampling

Suspended sediment was sampled during the long rainy seasons in 2018 and during the drier period of the start
of the long rainy season in 2019 (May-September 2018 and April-May 2019). The sediment sampling covered 20 and 12 sampling days in the natural forest, 22 and 13 days in the tea-tree plantation and 13 and 16 sampling days in the smallholder agriculture catchment in 2018 and in 2019, respectively. We deployed three different methods for suspended sediment sampling: time-integrated sampling with sediment traps ($n$=88) following the method by Phillips et al. (2000) (Figure 2d), manual ($n$=6) and automatic ($n$=7) (3700 Full-size portable
sampler, Teledyne ISCO, Lincoln, USA) storm event-based bulk sampling (Table 2). The manual event-based sampling was conducted next to the installed time-integrated samplers at each catchments outlet (Figure 1). Time-integrated samplers were emptied of accumulated suspended sediment after three to five days. The storm event-based samples were retrieved during a storm manually with bulk river water samples (~10 L). The auto-sampling was only conducted at the outlet of the smallholder agriculture catchment, whereby samples were
collected during the rising and falling stages of a storm event at 30 minute interval and composed to a bulk sample. Sediment in suspension from all three sampling methods was extracted through settling and sedimentation followed by air-drying in aluminium trays.

**Table 2 Total number of samples of time-integrated, manual and automatic storm event-based sediment sampling of the natural forest, tea-tree plantations and smallholder agriculture catchments in 2018 and 2019.**

| Catchment | Year | Number of time-integrated sediment samples | Number of manual storm event-based bulk samples | Number of automatic storm event-based bulk samples |
|---|---|---|---|---|
| **Natural forest** | 2018 | 20 | - | - |
| | 2019 | 12 | 2 | - |
| **Tea-tree plantations** | 2018 | 22 | - | - |
| | 2019 | 11 | 3 | - |
| **Smallholder agriculture** | 2018 | 13 | - | - |
| | 2019 | 10 | 1 | 7 |




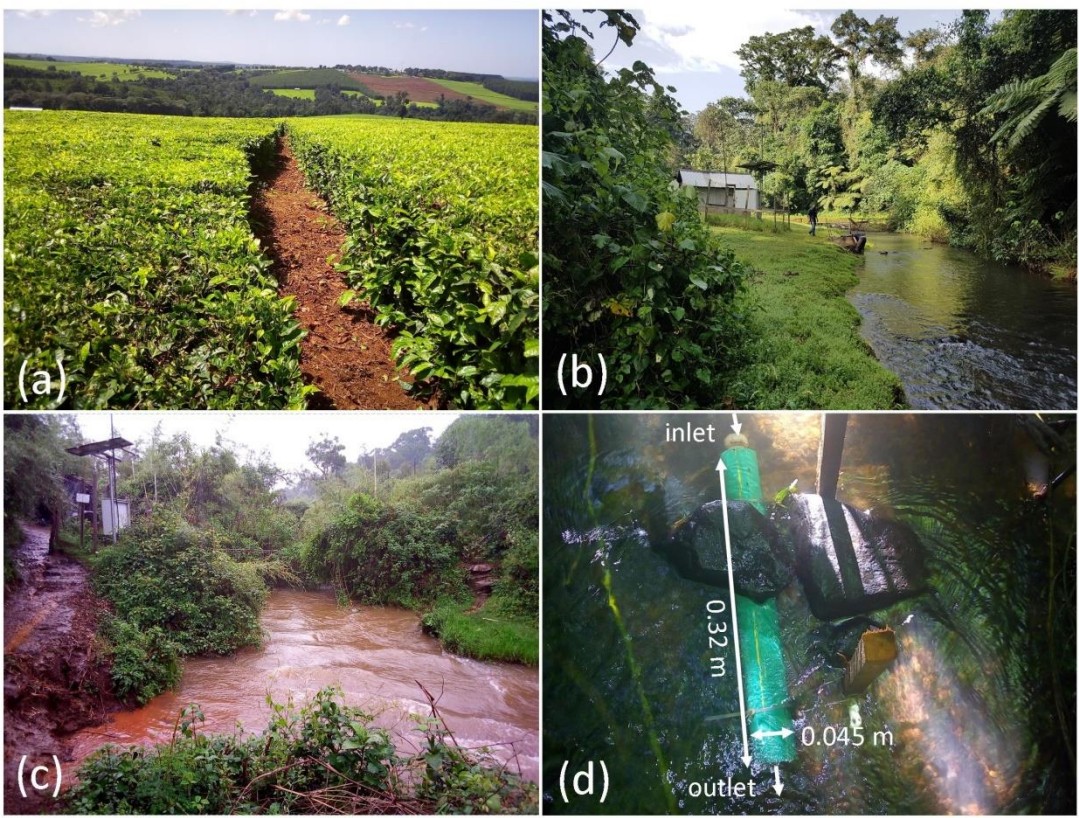

**Figure 2 (a) Tea-tree plantation catchment, (b) outlet of the natural forest and (c) outlet of the smallholder agriculture catchment and (d) time-integrated sediment trap.**

### 2.5 Processing and qualitative analysis of suspended sediment

An aliquot of each sediment sample (>230 mg) was ground using a ball mill grinder for further laboratory analysis. The ground samples were analysed for total carbon (TC), total nitrogen (TN) and total phosphorus (TP) concentrations. In this context, TC, TN and TP refer to the total C, N and P concentration in suspended sediments, which corresponds to particulate C, N and P (expressed in $g\,kg^{-1}$). For TC and TN concentration measurements, a sub-sample of 30 mg of ground sediment was wrapped in tin capsules and combusted in an

elemental micro-analyser (Elementar vario EL III, Elementar Analysensysteme GmbH, Langenselbold, Germany) at 950°C. For TP concentration measurements, a sub-sample of 200 mg of ground sediment was digested in 4.4 ml of sulphuric acid-hydrogen peroxide digest reagent and heated to 400°C for two hours (Allen

et al., 1974). After diluting the digestate twice, TP was determined using colorimetry based on a reaction with acidic molybdate in the presence of antimony which forms an antimony-phosphomolybdate complex. Ascorbic

acid turns the complex in an intensely blue (phosphomolybdenum blue) which is measured spectrophotometrically at 880 nm in a segmented flow analyser (Auto-analyser 3HQ, SEAL Analytical Ltd., Hampshire, United Kingdom). The remaining sediment samples were analysed for organic matter using gravimetric weight change by loss on ignition at 500°C for 4 hours in a muffle furnace.

To estimate nutrients and carbon concentrations in stream water (mg L$^{-1}$), the mean suspended sediment

concentrations (mg L$^{-1}$) for each sampling period were multiplied with the particulate nutrients and carbon concentrations (g kg$^{-1}$) per dry weight sediments. For TC, TN and TP load calculations, the mean discharge and suspended sediment concentration was obtained for each sediment sampling period. The concentrations of TC, TN and TP (g kg$^{-1}$) were multiplied to the mean discharge (m$^3$ s$^{-1}$) and mean suspended sediment concentrations (mg L$^{-1}$) to obtain the sediment-associated loads (t day$^{-1}$). Annual suspended sediment-associated TC, TN and

TP yields were calculated by integrating the mean of the daily loads to annual loads and relating it to the catchment area.

### 2.6   Data analysis

All data were analysed for normality using the Shapiro-Wilk test. We tested significant differences on particulate nutrients and total carbon values among the different land uses between the two years using the

Kruskal-Wallis test for analyses of variances. The effect of land use on particulate nutrients and total carbon within each year were tested for significance using the pairwise Wilcoxon rank sum test. To test the linear relationship between two macroelements, the correlation coefficient r was identified between each two of the macroelements. All significant differences reported are at $p < 0.05$.

## 3   Results

### 3.1   Hydrological and suspended sediment responses

Mean annual rainfall for the study period (2018-2019) was 1,989, 2,006, and 1,671 mm yr$^{-1}$ for the natural forest, tea-tree plantation and smallholder agriculture catchments, respectively. In the natural forest and the tea-tree plantations, 2019 was wetter than 2018, while the smallholder agriculture catchment was drier in 2019.





Mean annual specific discharge was highest in the natural forest catchment followed by the tea-tree plantation

and the smallholder agriculture catchments with 806 (778-834) mm yr$^{-1}$, 678 (642-714) mm yr$^{-1}$ and 658

(634-682) mm yr$^{-1}$. The catchment runoff coefficient was highest for the natural forest catchment (0.41) and

smaller for the tea-tree plantations and the smallholder agriculture with a mean of 0.34 and 0.39, respectively.

The mean annual suspended sediment yield for the two years was highest in the smallholder agriculture with

231 (215-248) t km$^{-2}$yr$^{-1}$ followed by the natural forest with 57 (54-62) t km$^{-2}$yr$^{-1}$ and the tea-tree plantation

240    catchments with 48 (44-52) t km$^{-2}$yr$^{-1}$ (Table 3).

**Table 3 Hydrological characteristics and total suspended sediment (TSS) (and 95 % confidence interval) of the three catchments under different land use natural forest (NF), tea-tree plantations (TTP) and smallholder agriculture (SHA) in the South-West Mau, Kenya of 2018 and 2019.**

| Site | Year | Annual rainfall [mm yr$^{-1}$] | Annual specific discharge [mm yr$^{-1}$] | Runoff coefficient[a] | TSS load [t yr$^{-1}$] | TSS yield [t km$^{-2}$ yr$^{-1}$] |
|---|---|---|---|---|---|---|
| NF | 2018 | 1,881 | 814 (783-845) | 0.43 (0.42-0.45) | 1,343 (1,230-1,461) | 37 (34-41) |
| | 2019 | 2,098 | 798 (772-823) | 0.38 (0.37-0.39) | 2,730 (1,925-2,234) | 76 (73-84) |
| | **Mean** | 1,989 | 806 (778-834) | 0.41 (0.39-0.42) | 2,037 (1,925-2,234) | 57 (54-62) |
| TTP | 2018[b] | 1,922 | 677 (637-717) | 0.35 (0.33-0.37) | 1,067 (972-1,167) | 32 (29-35) |
| | 2019 | 2,089 | 679 (647-711) | 0.32 (0.31-0.34) | 2,105 (1,933-2,284) | 63 (58-69) |
| | **Mean** | 2,006 | 678 (642-714) | 0.34 (0.32-0.36) | 1,586 (1,453-1,725) | 48 (44-52) |
| SHA | 2018 | 1,870 | 942 (909-974) | 0.50 (0.49-0.52) | 5,244 (4,872-5,629) | 193 (179-207) |
| | 2019[c] | 1,473 | 375 (359-390) | 0.25 (0.24-0.27) | 7,336 (6,835-7,854) | 270 (251-289) |
| | **Mean** | 1,671 | 658 (634-682) | 0.39 (0.38-0.41) | 6,290 (5,853-6,741) | 231 (215-248) |

[a]Specific discharge as proportion of annual rainfall
245    [b]Gaps in discharge and suspended sediment data due to sensor malfunctioning
[c]Gaps in discharge and suspended sediment data due to theft of power supply

In all three catchments, discharge followed the rainfall pattern. The rising limb of the hydrograph was generally

steep, followed by either steep or gentle falling limbs, depending on the magnitude of the storm event.

Discharge peaked during the long rainy season between April and July in 2018. In contrast, 2019 experienced a

250    delayed onset of the rains and the highest discharge peaks occurred between October and December. Suspended

sediment peaks followed the same pattern as discharge (Figure 3).







**Figure 3 Daily accumulated rainfall (R) [mm day$^{-1}$], specific discharge (Q) [mm day$^{-1}$] and suspended sediment yield (SSY) [t km$^{-2}$ day$^{-1}$] aggregated from 10 minute resolution with 95 % confidence interval (CI) of the (a) natural forest, (b) smallholder agriculture and (c) tea-tree plantation catchments in the South-West Mau, Kenya between January 2018 and December 2019.**

### 3.2    TC, TN and TP concentrations on sediment and in the stream

Suspended sediment sampling for the particulate nutrients and carbon analysis was conducted during the high flows in 2018 (May-September), while in 2019 the sampling coincided with the start of the long rainy season during April and May (Figure 3). The TC concentrations were 7 % lower in the natural forest catchment and 15 % higher in the smallholder agriculture in 2019 than in the previous year. The TN concentrations were 8-20 % higher in 2019 than in 2018 for all three catchments. In the natural forest and the tea-tree plantation catchments, the TP concentrations were 25 % and 17 % higher, respectively, and 38 % lower in the smallholder agriculture catchment in 2019 than in 2018 (Figure 4).

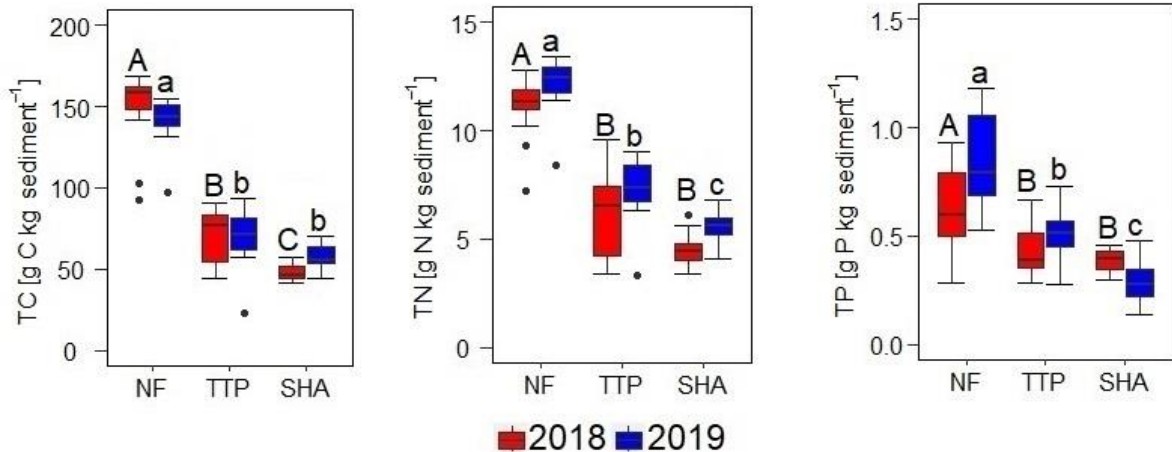

**Figure 4 Particulate TC, TN and TP concentrations [g kg sediment$^{-1}$] of time-integrated, manual and automatic collected samples of the natural forest (NF), tea-tree plantation (TTP) and smallholder agriculture (SHA) catchments in the South-West Mau, Kenya in 2018 and 2019. Different letters indicate significant differences between land uses (p<0.05).**

Particulate TC, TN and TP concentrations were significantly higher in the natural forest catchment than in the tea-tree plantation and smallholder agriculture catchments in both years. The TC concentrations were significantly higher in the tea-tree plantations than in the smallholder agriculture in 2018, while there was no difference in 2019. TN and TP concentrations were significantly higher in the tea-tree plantation than in the

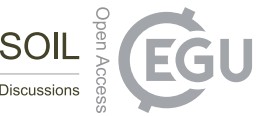

smallholder agriculture catchment in 2019, but concentrations were not significantly different between the
tea-tree plantations and the smallholder agriculture in 2018 (Table 4).

**Table 4 Mean ± standard deviation of particulate TC, TN and TP concentrations [g kg sediment$^{-1}$] of time-integrated, manual and automatically collected samples, suspended sediment concentrations (TSS) [mg L$^{-1}$] and TC, TN [mg L$^{-1}$] and TP [µg L$^{-1}$] concentrations in water at the outlet of the natural forest (NF), tea-tree plantations (TTP) and the smallholder agriculture (SHA) in the South-West Mau, Kenya based on 13-22 sampling days for the sampling campaign from May-October 2018 and 14-18 sampling days for the period April-June 2019.**

| Site | Year | Concentrations in sediment | | | Concentrations in stream water | | | |
|------|------|-----------|-----------|-----------|-----------|-----------|-----------|-----------|
| | | TC [g kg$^{-1}$] | TN [g kg$^{-1}$] | TP [g kg$^{-1}$] | TSS [mg L$^{-1}$] | TC [mg L$^{-1}$] | TN [mg L$^{-1}$] | TP [µg L$^{-1}$] |
| **NF** | 2018[a] | 151.0±20.5 | 11.2±1.3 | 0.6±0.2 | 43.6±10.9 | 6.6±1.8 | 0.5±0.1 | 28.5±13.9 |
| | 2019[b] | 141.4±14.6 | 12.2±1.3 | 0.9±0.2 | 43.4±22.0 | 6.0±2.6 | 0.5±0.2 | 36.2±20.5 |
| | **Mean** | 146.2±17.6 | 11.7±1.3 | 0.7±0.2 | 43.5±16.4 | 6.3±2.2 | 0.5±0.2 | 32.4±17.2 |
| **TTP** | 2018[a] | 69.2±16.9 | 6.1±1.8 | 0.4±0.1 | 50.8±48.1 | 3.6±4.3 | 0.3±0.3 | 21.9±23.8 |
| | 2019[b] | 69.5±17.3 | 7.3±1.5 | 0.5±0.1 | 104.7±127.3 | 6.3±6.9 | 0.7±0.7 | 43.4±36.9 |
| | **Mean** | 69.4±17.1 | 6.7±1.6 | 0.5±0.1 | 77.8±87.7 | 5.0±5.6 | 0.5±0.5 | 32.7±30.4 |
| **SHA** | 2018[a] | 48.1±4.9 | 4.5±0.8 | 0.4±0.1 | 188.0±151.7 | 8.7±6.4 | 0.8±0.6 | 70.6±52.0 |
| | 2019[b] | 56.9±7.7 | 5.6±0.6 | 0.3±0.1 | 231.4±441.0 | 11.9±22.9 | 1.2±2.5 | 63.1±123.1 |
| | **Mean** | 52.5±6.3 | 5.0±0.7 | 0.3±0.1 | 209.7±296.4 | 10.3±14.7 | 1.0±1.5 | 66.8±87.6 |

[a]wet period May-October 2018
[b]drier period April-June 2019

The mean TC, TN and TP concentrations in the stream water for both years, estimated based on the particulate macronutrient concentrations and suspended sediment concentrations, were highest for the smallholder agriculture and lowest for the natural forest catchment and tea-tree plantations (Table 4).


The natural forest catchment had the highest percentage of organic matter in suspended sediments with 31 %, followed by the tea-tree plantations with 24 % and the lowest percentage was measured in the smallholder agriculture catchment with 16 %.

### 3.3    Stoichiometric C:N, C:P and N:P ratios and their relationships

The C:N ratio in the sediment from natural forest (12.6±1.0) was significantly higher ($p<0.05$) than those of the tea-tree plantations (10.5±1.2) and the smallholder agriculture (10.6±1.3) for both years, while the C:N ratio was not significantly different between the tea-tree plantations and the smallholder agriculture ($p>0.05$). The C:P



ratio in the natural forest (269.8±106.4) was significantly higher than the tea-tree plantations (160.5±39.2) and the smallholder agriculture (126.7±22.6) in 2018, but the C:P ratio was not significantly different between the

three catchments in 2019. The N:P ratio was significantly higher in the natural forest (19.9±6.9) than the smallholder agriculture (11.8±3.0) in 2018, while 2019 showed no significant difference between the three catchments (p>0.05). The C:N ratio was significantly higher in 2018 than in 2019 in the natural forest and the tea-tree plantations, while the C:P and N:P ratio was significantly higher in 2019 than 2018 in the smallholder agriculture (Figure 5). The sediment-associated C:N:P ratio was highest in the natural forest catchment with

225 C: 17N: 1P, followed by the smallholder agriculture with 172 C: 16N: 1P and the tea-tree plantations with 148 C:14 N: 1P, whereas the N:P ratio was similar for all three catchments.

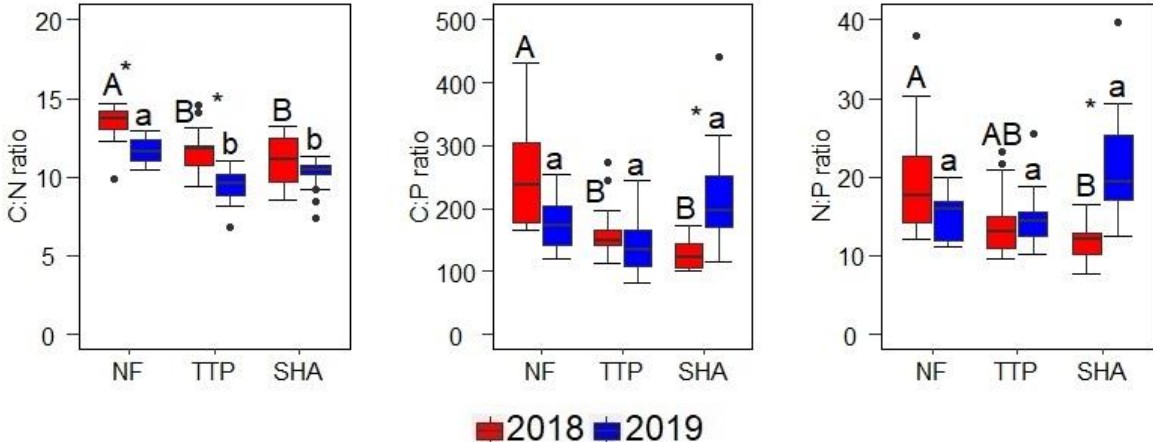

**Figure 5 Macronutrient ratios for carbon-nitrogen (C:N), carbon-phosphorus (C:P) and nitrogen-phosphorus (N:P) of the natural forest (NF), tea-tree plantation (TTP) and smallholder agriculture (SHA) catchments in the**
**South-West Mau, Kenya. Different letters indicate significant differences between catchments and the asterisk indicates significant differences within one catchment between years (p<0.05).**

TC, TN and TP concentrations were correlated against each other for all three catchments (Figure 6). The strongest relationship was observed between TC and TN within the tea-tree plantations (r=0.87), followed by the smallholder agriculture (r=0.71) and the natural forest catchment (r=0.64). The correlation between TN and

TP was strongest in the tea-tree plantations (r=0.58). No significant relationships were found between TC and TP in the natural forest and in the smallholder agriculture catchment with low correlation coefficient values ranging between -0.15 and 0.04, and between the correlation of TN and TP in the smallholder agriculture (r=-0.21) (p>0.05). The correlations of the natural forest catchment were in general higher than those of the tea-tree plantations and the smallholder agriculture (Figure 6).





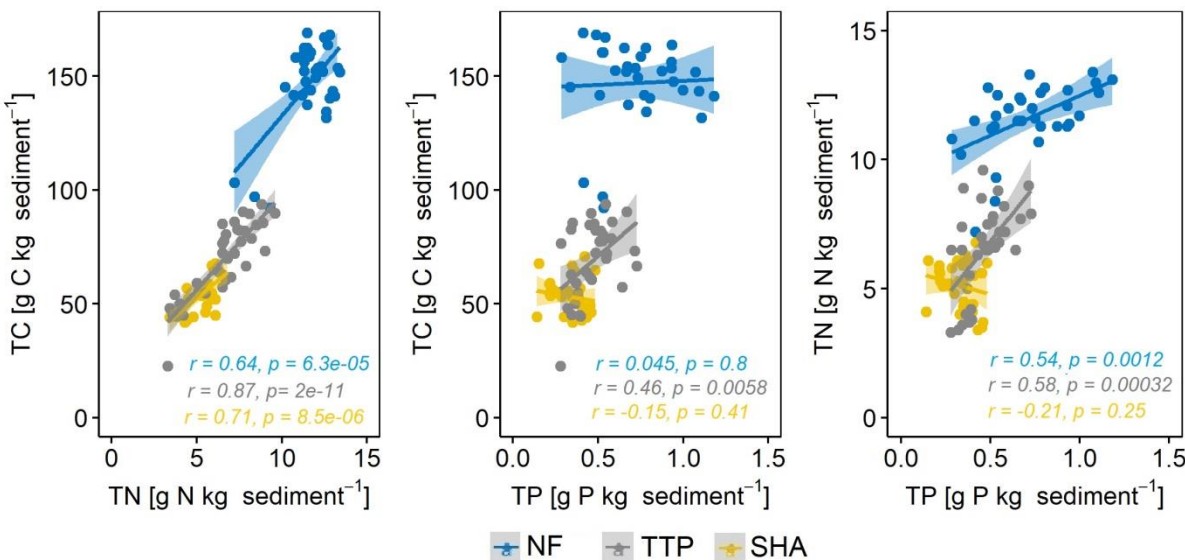

+ NF    + TTP    + SHA


**Figure 6 Correlations with correlation coefficient r obtained between total carbon (TC), total nitrogen (TN) and total phosphorus (TP) [g kg sediment$^{-1}$] concentrations of the natural forest (NF), the tea-tree plantation catchments (TTP) and the smallholder agriculture (SHA) in the South-West Mau, Kenya in 2018 and 2019. Significant difference at p<0.05.**

**3.4    Sediment-associated TC, TN and TP loads**

The mean daily suspended sediment loads for the sampling period were highest in the smallholder agriculture, followed by the natural forest and lowest in the tea-tree plantations (14.6±15.0 t day$^{-1}$, 4.9±2.5 t day$^{-1}$ and 3.4±3.9 t day$^{-1}$, respectively). The mean daily TC and TN load in suspended sediment during the sampling periods in 2018 and 2019 was highest for the smallholder agriculture (668.9±661.9 kg day$^{-1}$ and

60.5±59.5 kg day$^{-1}$), followed by the natural forest (496.9±335.3 kg day$^{-1}$ and 37.2±20.9 kg day$^{-1}$) and the tea-tree plantation catchment (193.4±178.6 kg day$^{-1}$ and 18.1±18.5 kg day$^{-1}$). For the sediment-associated TP, the highest daily loads were observed in the smallholder agriculture followed by the natural forest and the tea-tree plantations (5.5±5.1 kg day$^{-1}$, 1.9±1.3 kg day$^{-1}$, and 1.4±1.3 kg day$^{-1}$, respectively). The mean annual sediment-associated TC, TN and TP yields of the sampling period is estimated to be highest for the smallholder

agriculture (9.0±8.9 t TC km$^{-2}$ yr$^{-1}$, 0.8±0.8 t TN km$^{-2}$ yr$^{-1}$ and 73.9±68.9 kg TP km$^{-2}$ yr$^{-1}$), followed by the natural forest and lowest for the tea-tree plantations (2.1±2.0 t TC km$^{-2}$ yr$^{-1}$, 0.2±0.2 t TN km$^{-2}$ yr$^{-1}$ and 14.9±13.6 kg TP km$^{-2}$ yr$^{-1}$) (Table 5).



**Table 5 Overview of total suspended sediment-associated (TSS) total carbon (TC), total nitrogen (TN) [t km$^{-2}$ yr$^{-1}$] and total phosphorus (TP) yields [kg km$^{-2}$ yr$^{-1}$] and total suspended sediment yields [t km$^{-2}$ yr$^{-1}$] based on 13-22 sampling days for the sampling campaign from May-October 2018 and 14-18 sampling days for the period April-June 2019.**

| Site | Year | TSS TC yield [t km$^{-2}$ yr$^{-1}$] | TSS TN yield [t km$^{-2}$ yr$^{-1}$] | TSS TP yield [kg km$^{-2}$ yr$^{-1}$] | TSS yield [t km$^{-2}$ yr$^{-1}$] |
|---|---|---|---|---|---|
| NF | 2018 wet period | 9.0±5.6 | 0.7±0.4 | 32.1±17.4 | 59.3±41.1 |
| | 2019 drier period | 1.2±1.2 | 0.1±0.1 | 7.0±8.6 | 8.5±9.3 |
| | **Mean** | 5.1±3.4 | 0.4±0.2 | 19.6±13.0 | 50.2±25.2 |
| TTP | 2018 wet period | 3.0±2.6 | 0.3±0.2 | 20.5±15.5 | 51.3±44.4 |
| | 2019 drier period | 1.2±1.3 | 0.1±0.2 | 9.3±11.7 | 24.0±41.9 |
| | **Mean** | 2.1±2.0 | 0.2±0.2 | 14.9±13.6 | 37.7±43.2 |
| SHA | 2018 wet period | 17.5±16.9 | 1.6±1.5 | 145.6±133.2 | 383.7±385.6 |
| | 2019 drier period | 0.4±0.8 | 0.05±0.1 | 2.3±4.5 | 8.2±15.9 |
| | **Mean** | 9.0±8.9 | 0.8±0.8 | 73.9±68.9 | 196.0±200.8 |

## 4 Discussion

### 4.1 Land use affects sediment-associated carbon and nutrients concentrations

This study shows that land use is a key control of suspended sediment-associated nutrients and carbon concentrations in the headwaters of the Sondu River Basin. Most significantly, and despite the application of fertilizer to the tea-tree plantation and smallholder agriculture catchments, the natural forest catchment has much higher particulate TC, TN and TP concentrations than the agricultural catchments. These results suggest that the sediment from the forest catchment is comprised of organic material with a high C:N:P ratio either originating from the forest floor or falling directly into the river. In contrast, sediments from the agricultural catchments are mix of mineral and organic matter and fertilizer additions do not balance with the loss of carbon, nitrogen and phosphorus from the system. This refutes our hypothesis that particulate-bound nutrients and carbon concentrations in sediment are highest in agricultural catchments where the majority of suspended sediment (77 %) originating from agricultural land (Stenfert Kroese et al., 2020a). Although concentrations of TC, TN and TP associated with sediments were lower in the smallholder agriculture catchment, because of higher sediment loads from the smallholder agriculture catchment the total sediment-associated loads of TC, TN and TP were higher. In addition, higher sediment concentrations in the smallholder catchment meant that TC, TN and TP concentrations in the stream water were also higher from that catchment (Table 4, Table 5).



The low C and nutrient concentrations in sediments from agricultural soils are associated with soils much lower in nutrients and organic matter compared to nutrient-rich forest soils. The native forest vegetation was converted to smallholder farms during the last few decades (Brandt et al., 2018), conversion that leads to reduced organic inputs and increased decomposition rates of soil organic matter following land management practices (Freibauer et al., 2004). Globally, an average decrease in soil C of around 30 % was found for soils after conversion of forests to croplands (Don et al., 2011; Murty et al., 2002), while a study of a converted native forest in the highlands of western Kenya showed a decrease by 30-40 % within the first 39 years after conversion (Nyberg et al., 2012). Similarly, a decline in soil organic carbon and nutrients was observed following conversion to agricultural cultivation in the same catchments of the Mau Forest Complex (Arias-Navarro et al., 2017; Owuor et al., 2018; Wanyama et al., 2018). The highest TC and TN concentrations ($81.1\pm24.2$ g kg$^{-1}$ and $4.9\pm2.3$ g kg$^{-1}$, respectively) were measured in the surface soil (0-0.05 m) of the natural forest catchment, compared to lowest concentrations on smallholder croplands ($56.9\pm11.1$ g kg$^{-1}$ and $2.1\pm1.2$ g kg$^{-1}$, respectively) (Owuor et al., 2018). Chiti et al. (2018) found a significant decline in soil organic carbon as a consequence of forest degradation in the same tropical montane forest. In their study, the organic horizon, the litter layer, had the highest TC concentration under primary forest ($412.3\pm23.2$ g kg$^{-1}$) compared to a degraded forest ($408.2\pm21.3$ g kg$^{-1}$) and to cypress ($398.6\pm19.6$ g kg$^{-1}$) and tea plantations ($381.7\pm17.6$ g kg$^{-1}$) that replaced the forest. With the reduction in the organic horizon a decline in soil TC was found in the uppermost mineral soil layer (<0.05 m) (Chiti et al., 2018). The suspended sediment is of mineral origin in the agricultural catchments and of organic origin in the natural forest catchment, which is reflected in the differences in organic matter and the C concentrations in suspended sediment (Table 4).

The higher C:N:P ratios and the relationship between TC and TN in the natural forest catchment (Figure 6) further imply that TN is of organic origin, where a lower and similar relationship in both agricultural catchments suggests a mix of mineral and organic sediment sources. A strong relationship was observed between TC and TP for the agricultural catchments, which may indicate that the mineralization of soil organic matter contributes to the available phosphorous, as observed in the study of Maranguit et al. (2017). Similarly, the study of Johnson et al. (2017) in a forested catchment in the Piedmont region, Maryland (USA) showed that sediment sources from forest floor litter had the highest TC and TN concentrations compared to near-stream sources such as stream bed and stream banks. The fresher material from the forest floor was likely the least degraded resulting in higher C and nutrient concentrations (Johnson et al., 2018). Old growth forests maintain a tight nutrient cycle through high and diverse aboveground biomass. They accumulate much more organic matter and





nutrients by decomposition of fresh litter material and production of humus than managed land use types where carbon inputs are smaller and losses larger (Dawson and Smith, 2007).

When compared with other monitored catchments, both within and outside the tropics the sediments in the forested catchment were more enriched with carbon, similar for nitrogen yet depleted in phosphorus, while the tea-tree plantation and smallholder agriculture catchments carbon and nitrogen fell within the range reported in other studies but phosphorus concentrations were lower (Table 6). The TC concentrations of the natural forest catchment of our study were more than twice the concentrations recorded in the Congo basin under disturbed

and undisturbed forest cover (14-95 %) (Coynel et al., 2005) and of disturbed agricultural catchments in temperate regions (López-Tarazón et al., 2016; Walling et al., 2001). Other catchments of disturbed tropical basins under mixed land use (Tana River basin, Kenya and Ayeyarwady and Thanlwin River, Myanmar) had lower TC concentrations (Bird et al., 2008; Tamooh et al., 2012) than the concentrations of all three catchments of this study.

The TN concentrations of the natural forest catchment were in the same range as those of sub-catchments under mixed land use of the Yangtze River, Jialing River and Wujian River, China, during the wet season, but exceeded the concentrations of intensified agricultural catchments in the USA, New Zealand and Spain by up to 12-fold (López-Tarazón et al., 2016; McDaniel et al., 2009; McDowell, 2015). The TN concentrations of the agricultural catchments of this study were within the ranges of concentrations in intensified temperate

agricultural catchments (Pavanelli and Selli, 2013; Walling et al., 2001) (Table 6).

The particulate TP concentrations of the natural forest, tea-tree plantations and smallholder agriculture catchments were lower than those reported for a tobacco cultivated catchment in sub-tropical Brazil, with a TP range of 0.09-3.58 g P kg$^{-1}$, with a similar mean annual rainfall of 1,938 mm yr$^{-1}$ (Bender et al., 2018). TP concentrations for agricultural catchments of temperate regions (Neal et al., 2006; Ramos et al., 2015;

Sandström et al., 2020) and catchments in China during the wet season (Wang et al., 2015) exceeded the concentrations of our study (Table 6).





**Table 6 Overview of particulate total carbon (TC), total nitrogen (TN) and total phosphorus (TP) mean concentrations [g kg⁻¹] of catchment studies around the world. Analysed sample material: SS = suspended sediment, BS = riverbed sediment and WS = water sample. DR Congo = Democratic Republic of Congo, TRPR = Tana River Primate Reserve.**

| Catchment/ basin | Country | Area [km²] | Land use | Sample | Study period [year][b] | Rainfall [mm] | TC [g kg⁻¹] | TN [g kg⁻¹] | TP [g kg⁻¹] | Reference |
|---|---|---|---|---|---|---|---|---|---|---|
| **Tropical catchments** | | | | | | | | | | |
| **Sondu** | Kenya | 36 | Forest | SS | 2018-2019 | 1,989 | 147.61 | 12.06 | 0.81 | This study |
| **Sondu** | Kenya | 33 | Agriculture | SS | 2018-2019 | 2,006 | 83.72 | 8.00 | 0.54 | This study |
| **Sondu** | Kenya | 27 | Agriculture | SS | 2018-2019 | 1,671 | 53.29 | 5.02 | 0.33 | This study |
| **Kora (Tana River)** | Kenya | 22,080 | Mixed land use | SS | 2009-2011 | 450-900 | 25.32[c] | n.a. | n.a. | Tamooh et al. (2014) |
| **Garissa (Tana River)** | Kenya | 32,500 | Mixed land use | SS | 2009-2011 | 450-900 | 19.92[c] | n.a. | n.a. | |
| **TRPR (Tana River)** | Kenya | 66,500 | Mixed land use | SS | 2009-2011 | 450-900 | 17.29[c] | n.a. | n.a. | |
| **Oubangui (Congo)** | DR Congo | 489,000 | Forest (22 %) | SS | 1990-1996 | 1,550 | 60.61 | n.a. | n.a. | Coynel et al. (2005) |
| **Mpoko (Congo)** | DR Congo | 23,900 | Forest (14 %) | SS | 1991-1994 | 1,550 | 41.67 | n.a. | n.a. | |
| **Ngoko-Sangha (Congo)** | DR Congo | 67,000 | Forest (95 %) | SS | 1991 | 1,550 | 61.17 | n.a. | n.a. | |
| **Congo/Zaire (Congo)** | DR Congo | 3,500,000 | Forest (50 %) | SS | 1990-1993 | 1,550 | 64.64 | n.a. | n.a. | |
| **Arroio Lajeado Ferreira** | Brazil | 1.2 | Mixed land use | WS | 2011-2015 | 1,938 | n.a. | n.a. | 1.22 | Bender et al. (2018) |
| **Streams in New Zealand** | New Zealand | <20,000 | Agriculture | BS | 2012 (02-03) | n.a. | 2.10 | 0.21 | 0.42 | McDowell (2015) |
| **Ayeyarwady and Thanlwin River** | Myanmar | n.a. | Mixed land use | SS | 2006 (05,08,09) | ~3,000 | 14.65[c] | n.a. | n.a. | Bird et al. (2008) |
| **Temperate and Mediterranean catchments** | | | | | | | | | | |
| **Duck Creek, Fox River, Wolf River** | USA | <9,666 | Mixed land use | BS | 2016-2017 | 748-800 | n.a. | n.a. | 0.47[d] | Kreiling et al. (2019) |
| **Embarras and Vermilion** | USA | <839 | Agriculture | BS | 2004 | 1,270 | n.a. | n.a. | 0.34[d] | 0.36[d] | McDaniel et al. (2009) |
| **Wye, Welland, Avon** | UK | 0.4-9.9 | Agriculture (more intensive) | BS | 2005-2006 | 671-905 | n.a. | n.a. | 1.56 | Palmer-Felgate et al. (2009) |
| **Wye, Avon** | UK | 0.4-9.9 | Agriculture (less intensive) | BS | 2005-2006 | 671-905 | n.a. | n.a. | 0.63 | |
| **Severn, Avon, Eve, Dart** | UK | <6,850 | Agriculture | SS | 1995-1996 | 600-2,300 | 53.70[c] | 4.47 | 1.49 | Walling et al. (2001) |
| **Upper Thames River** | UK | 3,500 | Mixed land use | WS | 1997 (1-6 years) | n.a. | n.a. | n.a. | 2.16-6.87 | Neal et al. (2006) |
| **Enxoé River** | Portugal | 61 | Agriculture | SS | 2010-2013 | 500 | n.a. | n.a. | 4.30 | Ramos et al. (2015) |
| **Ésera** | Spain | 1,484 | Mixed land use | SS | 2011-2012 | 1,069 | 60.81 | 0.80 | n.a. | López-Tarazón et al. (2016) |
| **Reno River** | Italy | 389 | Agriculture | SS | 2000-2009 (flood events) | 950-1,015 | n.a. | 9.50 | 0.46 | Pavanelli and Selli (2013) |
| **South Sweden** | Sweden | <33.1 | Agriculture (54-59 %) | WS | 2004-2017 | 539-623 | n.a. | n.a. | 1.07 | Sandström et al. (2020) |
| | | <16.3 | Agriculture (89-93 %) | WS | 2004-2017 | 506-709 | n.a. | n.a. | 1.78 | |
| **Yangtze River** | China | n.a. | Mixed land use | SS | 2010 (05-10) | n.a. | n.a. | 1.23 | 0.54 | Wang et al. (2015) |
| | | | | | 2011 (11-04) | n.a. | n.a. | 11.36 | 4.09 | |
| **Jialing River** | China | n.a. | Mixed land use | SS | 2010 (05-10) | n.a. | n.a. | 3.29 | 0.55 | |
| | | | | | 2011 (11-04) | n.a. | n.a. | 22.50 | 5.00 | |
| **Wujian River** | China | n.a. | Mixed land use | SS | 2010 (05-10) | n.a. | n.a. | 4.09 | 0.86 | |
| | | | | | 2011 (11-04) | n.a. | n.a. | 18.18 | 4.55 | |

n.a. = not available
[b]month of the year
[c]particle organic carbon
[d]concentrations presented in median



## 4.2    Implications of nutrient losses from agricultural soils

The low sediment-bound concentrations (TC, TN and TP) in the smallholder agriculture catchment reflect soils with much lower organic matter and nutrients than the forest due to intensive cropping without appropriate practices for restoration of soil fertility. In addition to the decline in organic matter after conversion, soils cultivated on steep hillslopes with poor soil conservation practices and excessive surface runoff are prone to erosion and further nutrient depletion (Stenfert Kroese et al., 2020b). This may have been intensified by the cultivation of a major nutrient miner, *Pyrethrum* daisy (*Chrysanthemum cinerariifolium*), following the clearance of the South-West Mau, which is known to promote surface soil erosion due to poor soil cover (Smaling et al., 1993). Similar soil nutrient losses have been observed in other densely populated tropical agricultural regions cultivated on steep hillslopes in Uganda (Lederer et al., 2015), Tigray (Ethiopia) (Girmay et al., 2009) or Kisii District (western Kenya) (Smaling et al., 1993) through nutrient losses, caused in particular by surface erosion and insufficient use of fertilizer application. Scarcity of land leads to an overexploitation of soil nutrients on agricultural land due to the absence of fallow periods.

There is a need for soil management practices to be focused on the retention of soil organic matter through boosting agricultural productivity, mulching or cover crops in the agricultural catchments to maintain soil fertility. Increased retention of crop residues creates a positive feedback loop for the accumulation of organic matter (Nyberg et al., 2012). Incorporating crop residues into the soil is challenging for smallholder farmers, who prefer to feed them to their livestock (Castellanos-Navarrete et al., 2015). Other management practices such as vegetative buffer strips, erosion ditches or *fanya juu* terracing can prevent soil erosion (Conelly and Chaiken, 2000; Tiffen et al., 1994). Improved soil stability through soil organic matter will reduce soil erosion and suspended sediments and the further loss of soil organic carbon and nutrients, and therefore integrative practices that boost the production biomass and reduce bare soil are required.

## 5    Conclusions

This study conducted in catchments in the headwaters of the Sondu River Basin of the South-West Mau, Kenya, shows that cultivated land uses have led to a pronounced decline in sediment-associated carbon and nutrient concentrations compared to native forest. The high carbon and nitrogen ratio in the natural forest catchment reveals that most particulate nitrogen is of organic origin. The native forest is rich in biomass, with a tighter nutrient cycle and likely high organic matter inputs. This is in contrast to the smaller nutrient stocks in the agricultural land, reflecting nutrient losses and the impoverishment of agricultural soils. The disturbance of soil through agricultural practices increases soil mineralization of soil organic matter, echoed in the lower organic matter content in suspended sediments. Despite the lower carbon and nutrient concentrations, the smallholder agriculture had the greatest particulate carbon and nutrient yields due to the higher export of suspended sediment yields passing the outlet. Elevated carbon and nutrient concentrations in sediments and in stream water contribute

to nutrient pollution and eutrophication impacting the downstream reaches such as Lake Victoria. Management practices should focus on retaining and storing soil organic matter of agricultural topsoils to increase soil organic carbon, and soil nutrients. Practices to manage soil organic matter should also control soil erosion and consequently reduce suspended sediment concentrations and the loss of nutrient-rich topsoil.

**Data availability**

The raw data are available online https://doi.org/10.17635/lancaster/researchdata/387 hosted by Lancaster University, United Kingdom.

**Author contribution**

JSK, MCR and JNQ designed the research. JSK conducted sampling, laboratory work and analysed results. JSK prepared the manuscript with input from MCR, JNQ, LB and SRJ.

**Competing interest**

The authors declare that they have no conflict of interest.

**Acknowledgements**

We thank the German Federal Ministry for Economic Cooperation and Development (Grant 81206682 "The Water Towers of East Africa: policies and practices for enhancing co-benefits from joint forest and water conservation") and the German 460 Science Foundation (Deutsche Forschungsgemeinschaft DFG, Grant BR2238/23-1) for providing financial support for this research. This work was also partially funded by the CGIAR program on Forest, Trees and Agroforestry led by the Centre for International Forestry Research (CIFOR). We would like to thank the tea companies, the Kenya Forest Service (KFS) and the chief of the smallholder agriculture catchment (Kuresoi sub-location) for supporting our research activities, and Megan Tomlinson for assisting with field work in 2018.

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
