# Peer review of "Particulate macronutrient exports from tropical African montane catchments point to the impoverishment of agricultural soils"

_SOIL, 2020_

## Referee Comment (RC1) · Anonymous Referee #1 · 16 Dec 2020

This study investigates suspended sediment and particle-bound nutrient fluxes from three catchments (with surfaces of $\sim$30 km$^2$) covered with different land uses in Kenya, East Africa. Hydro-sedimentary monitoring was conducted at the outlet during 2 years. This manuscript is very well written, documented and illustrated (figures and tables are very well done), and the research topic fits with the scope of SOIL journal. In my opinion, minor to moderate revisions should be required before the final acceptance of the manuscript. Detailed comments are provided below.

Abstract The quality of the abstract writing could be improved in my opinion (the quality of this section is not as good as the rest of the manuscript). L. 14 "catchments generate high concentrations of suspended sediment" » should be rephrased L.17 "tightly connected to processes" » unclear, please rephrase L.19 "with widespread land conversion" » maybe specify the type of conversion of interest here L.21 unclear what you mean with the "knowledge base" here LL.23-24 maybe add the corresponding catchment surface areas here L.27 not sure "tighter" is the right term to use here?

Introduction L.34 could you specify what you refer to as "high" here? L.63 "sediment-associated nutrients" » which exact parameter are you referring to here? LL.65-67 were these different interpretations obtained in different contexts/environments?

Materials and Methods L.89 converted into...? L.95 I guess that based on this statement and the characteristics shown in Table 1, these 3 catchments are hypothesized to be similar in terms of slope, surface, soil type,... characteristics? Maybe state this explicitly? L.106 what do you consider to be "moderate to high amounts of organic matter"? Table 1: maybe add a category of characteristics to compare "signs/types of erosion" observed in the three catchments? For instance, on L. 134 in the text, you mention the occurrence of gullies. Are there other signs/types of erosion in the study areas? L.179 "long rainy season"» could you contextualize this better? Is it normal or not in this part of Kenya? What is "long"?

Results The text is really straight-to-the point and easy to read and to follow. It is clear that sediment fluxes are the highest from the agricultural catchment, although when I read the abstract, I had an opposite impression. Could you double-check that the text is not misleading on this point? Then, your results show that particle-bound nutrient concentrations are depleted in the agricultural catchment compared to the other catchments (in particular the forest catchment). Still, the nutrient fluxes from the agricultural catchment remain high (even higher than those from the other catchments, at least during the wet year, i.e. 2018; Table 5). Maybe it would be helpful to mention in the text (in % or in number of times) how higher/lower are the fluxes (either of sediment or of nutrients) when you compare the sites/years to contextualise this better. Regarding this topic, you focus in the text on the surface erosion processes, but what about

the occurrence of subsurface erosion processes in the investigated catchments? You mention the occurrence of gullies in the text, what about the potential contribution of landslide or channel bank erosion to sediment transiting these rivers? This subsoil material should be depleted in C/N/P, which may impact the fluxes exported from the catchments and your conclusions regarding management options.

Discussion L.339: "land use is a key control" » is it land use or land cover/management? Or both? LL.369-372: about the discrimination between mineral and organic origins: is there really such a dichotomy or can it be nuanced through the mobilization/transport/deposition of organo-mineral complexes? LL.395-400: nice to have compiled all the data shown in Table 6; of course, it is really valuable to compare your results with those found in similar/tropical environments. Just a random question: is it meaningful to compare these results with those found in Spain, for instance? Are these environments /land management modes comparable? L.424 : again, you refer explicitly to "surface erosion processes", but how can you convince the readers that subsurface erosion is negligible in these steep catchments?

---

## Referee Comment (RC2) · Anonymous Referee #2 · 6 Jan 2021

The paper by Stenfert Kroese et al. addresses the impact of land use on particulate carbon and nutrient export from tropical montane catchments in the South-West Mau in Kenya and shows that soil fertility is lost with the conversion from natural forest to cultivated land. This study fills in an important knowledge gap on particulate nutrient export of tropical ecosystems in East Africa. The manuscript is mostly well written and clearly structured. I recommend it for publication in SOIL after some revisions.

My main concern is the way the data is presented. In Figures 4 and 5 and Table 4, concentrations and ratios are presented for each sampling year separately. This might be prone to misinterpretation, as different seasons were sampled for each year. In

[Figure]

Table 5 it is stated that the data are from wet period (2018) and drier period (2019). I think it's necessary to be consistent throughout the manuscript, thus, the same labelling is necessary for Figures 4 and 5 and Table 4. Related to this: While in the results differences between years/seasons are acknowledged, in the discussion the whole seasonality is neglected. Especially the C:N, C:P, and N:P ratios seem to differ during the different seasons. Might this indicate different sediment sources? I think this needs to be addressed shortly in the discussion. Furthermore, the methods section lacks some more details: How where the data stored? Did you use an external datalogger or did the sensor log internally? The stealing of power supply and subsequent data loss is mentioned, but it is not clear how the setup was powered. How long did you let the sediments settle before air drying the aluminum trays?

Specific comments: Introduction p. 2 L52: What is the impact of an increased turbidity in streams? p. 3 L65ff: The presented results are from which ecosystems? All tropical? p. 3 L69: "This is an important knowledge gap" Methods p. 9 L176: Integrating missing discharge data linearly does not seem right, if rainfall data is available discharge can be correlated to rainfall? p. 9 L179/180: Instead of calling it the "drier period of the start of the long rainy season" maybe use "onset of the rainy season" p. 11 L212: organic matter content p. 11 L219: I'm wondering how representative the yearly yields of sediment-associated TC, TN and TP are if you use only data from 3 sampling periods over two years. Or did you relate the C and nutrient yields to the turbidity data? If so, it is not clearly described. p. 11 Data analysis: Which programs did you use for data analysis? Results p. 12 L235: State somewhere in the text that the values between the brackets are the 95% CI. p. 12 L236: Define the catchment runoff coefficient and how you calculated it in the text Figure 3: This is a really nice figure, however, it is not discussed at all in the manuscript. For example, I'm wondering why SSY are a lot higher during the 2019 season compared to the 2018 season in the NF and TTP catchment, although the discharge seems lower. I see that this has been more the focus of recent work by the authors and is not the aim of the present manuscript, however, in my opinion if the data is presented like this in the manuscript it should be

discussed accordingly. p. 15 L286-288: are the reported OM contents mean values over both years? Maybe add these values to a table? p. 17 L324-326: add C and N to the units to avoid confusion: kg C day-1 and kg N day-1. Also for TP in line 328. Section 3.4: Does it make sense to calculate mean annual yields of the sampling periods? How representative are these values? I think this needs to be addressed in the discussion. Discussion p. 19 L359: Do you know how long ago the conversion from forest to agricultural land occurred in your study sites? p. 19 L362: For an easier reading: put the values in the bracket for TC and TN concentrations behind "natural forest catchment". Also add C and N to the units. p. 19 L375: Figure 6 shows no strong correlation between TC and TP for SHA. Be precise that only TTP shows a significant relationship.

---

## Author Comment (AC1) · 15 Jan 2021

We would like to thank Reviewer #1 for the feedback provided to our manuscript. We will take their comments into account when revising the manuscript. Please find below the responses.

Anonymous Referee #1 This study investigates suspended sediment and particle-bound nutrient fluxes from three catchments (with surfaces of _30 km2) covered with different land uses in Kenya, East Africa. Hydro-sedimentary monitoring was conducted at the outlet during 2 years. This manuscript is very well written, documented and illustrated (figures and tables are very well done), and the research topic fits with the scope of SOIL journal. In my opinion, minor to moderate revisions should be required before the final acceptance of the manuscript. Detailed comments are provided below.

Abstract The quality of the abstract writing could be improved in my opinion (the quality of this section is not as good as the rest of the manuscript).

L. 14 "catchments generate high concentrations of suspended sediment" should be rephrased

Response: We amended the sentence to 'Agricultural catchments in the tropics often generate high concentrations of suspended sediments following the conversion of natural ecosystems' We will revisit the rest of the abstract and sharpen up the English.

L.17 "tightly connected to processes" unclear, please rephrase

Response: We amended to 'tightly connected to an increase in riverine particulate carbon and nutrient export'.

L.19 "with widespread land conversion" maybe specify the type of conversion of interest here

Response: We included 'with widespread land conversion from forests to agriculture'.

L.21 unclear what you mean with the "knowledge base" here

Response: We amended to 'In this study, we assess the effect of land use on particulate TC, TN and TP concentrations.'

LL.23-24 maybe add the corresponding catchment surface areas here

Response: We included the catchment areas: 'a natural montane forest (35.9 km2), a tea tree plantation (33.3 km2) and a smallholder agriculture (27.2 km2) catchment' (L. 23-24).

L.27 not sure "tighter" is the right term to use here?

Response: We believe that the term 'tighter nutrient cycle' is used in an appropriate way due to a fast mineralization and decomposition of organic matter and the input of fresh organic matter through a high and diverse aboveground biomass in natural forest ecosystems with little loss of nutrients from the system.

Introduction L.34 could you specify what you refer to as "high" here?

Response: To be more accurate we included 'These sediment concentrations can be particularly high (up to 8,387 t km-2 yr-1) in the steep highlands of East Africa . . .' and we amended the reference to 'Stenfert Kroese et al., 2020b; Vanmaercke et al., 2014)'.

L.63 "sediment-associated nutrients" Âż which exact parameter are you referring to here?

Response: We refer here to N and P we amended the sentence accordingly 'Sediment-associated nutrients (N and P) . . .'.

LL.65-67 were these different interpretations obtained in different contexts/environments?

Response: These interpretations were obtained from studies in temperate and tropical regions. We included 'Other studies in temperate and tropical regions . . .' and 'Walling et al. (1997) and Bender et al. (2018) observed that P loads mainly occur in particulate form in temperate and subtropical catchments, while N is mainly transported in dissolved form in a temperate catchment in China (Wang et al., 2015)'.

Materials and Methods L.89 converted into. . .?

Response: We corrected to 'converted into . . .'.

L.95 I guess that based on this statement and the characteristics shown in Table 1, these 3 catchments are hypothesized to be similar in terms of slope, surface, soil type,.

.. characteristics? Maybe state this explicitly?

Response: We amended the sentence to 'The study catchments were chosen based on the criteria of different land use: (1) natural forest (NF; 35.9 km2), (2) tea tree plantations (TTP; 33.3 km2) and (3) smallholder agriculture (SHA; 27.2 km2) and comparability between the catchment characteristics, such as surface area, morphology, geology, pedology, slope and climate (Figure 1 & Table 1)'.

L.106 what do you consider to be "moderate to high amounts of organic matter"?

Response: We included the percentage of organic matter: '... with moderate (15-30%) to high (>30%) amounts of organic matter'.

Table 1: maybe add a category of characteristics to compare "signs/types of erosion" observed in the three catchments? For instance, on L. 134 in the text, you mention the occurrence of gullies. Are there other signs/types of erosion in the study areas?

Response: Included a category 'Types of erosion' in Table 1.

L.179 "long rainy season"Âż could you contextualize this better? Is it normal or not in this part of Kenya? What is "long"?

Response: Kenya has a bimodal rainfall pattern with a long rainy season usually covering the months between March and June and a short rainy season from October to December. The rainfall pattern and the different seasons are already introduced in L. 100-103 in section 2.1 Catchment characteristics. 'The region has a bimodal rainfall pattern with a long rainy season (March June) and a short rainy season (October December) with a continued intermediate rainy season between the two wet seasons (July September). The driest months are in January and February.'

Results The text is really straight-to-the point and easy to read and to follow. It is clear that sediment fluxes are the highest from the agricultural catchment, although when I read the abstract, I had an opposite impression. Could you double-check that the text is not misleading on this point?

Response: When comparing the macronutrient concentrations between the catchments the natural forest catchment had the highest concentrations. However, because of higher sediment loads from the smallholder agriculture catchment, the total sediment-associated loads of the nutrients and carbon were higher compared to the natural forest catchment. We included a sentence in the abstract: 'Particulate carbon and nutrient concentrations were up to three fold higher (p<0.05) in the natural forest catchment compared to fertilized agricultural catchments. However, because of higher sediment loads from the smallholder agriculture catchment the total sediment associated loads of TC, TN and TP were higher compared to the natural forest and the tea-tree plantation catchment.'

Then, your results show that particle-bound nutrient concentrations are depleted in the agricultural catchment compared to the other catchments (in particular the forest catchment). Still, the nutrient fluxes from the agricultural catchment remain high (even higher than those from the other catchments, at least during the wet year, i.e. 2018; Table 5). Maybe it would be helpful to mention in the text (in % or in number of times) how higher/lower are the fluxes (either of sediment or of nutrients) when you compare the sites/years to contextualise this better.

Response: We amended the paragraph and included the number of times the smallholder agriculture catchment is higher compared to the natural forest and tea-tree plantations.

Regarding this topic, you focus in the text on the surface erosion processes, but what about the occurrence of subsurface erosion processes in the investigated catchments? You mention the occurrence of gullies in the text, what about the potential contribution of landslide or channel bank erosion to sediment transiting these rivers? This subsoil material should be depleted in C/N/P, which may impact the fluxes exported from the catchments and your conclusions regarding management options.

Response: Thank you for this comment. Sediment sources in the smallholder agriculture catchment are more diverse and originate with the greatest contribution from agricultural land but also from subsurface sources such as deeply incised unpaved tracks, gullies or channel banks as shown in a sediment fingerprinting study by Stenfert Kroese et al. 2020a (L. 354). Subsoil material is observed to be depleted in macronutrient concentrations. We therefore included a paragraph to highlight the occurrence of subsurface sources and their impact on depleted macronutrient concentrations: 'In the smallholder agriculture catchment, the lowered concentrations of sediment TC, TN and TP can be explained by sediment originating from the subsurface where nutrient concentrations are lower (Russell et al. 2001; Gellis et al. 2009; Wanyama et al. 2018). This was demonstrated by Stenfert Kroese et al. (2020a), in a sediment fingerprinting study, that the subsoil sources are of increased importance in the smallholder agriculture compared to the natural forest and the tea-tree plantation catchment due to exposure of subsoil to erosion processes.'

Discussion L.339: "land use is a key control" is it land use or land cover/management? Or both?

Response: We use the term 'land use' due to consistency throughout our study as we compared three catchments under distinct land use. However, the management of soil cover is certainly important and we no acknowledge this in 'this study shows that land use and management is a key control . . .'.

LL.369-372: about the discrimination between mineral and organic origins: is there really such a dichotomy or can it be nuanced through the mobilization/transport/deposition of organo-mineral complexes?

Response: We removed the sentence 'The suspended sediment is of mineral origin in the agricultural catchments and of organic origin in the natural forest catchment, which is reflected in the differences in organic matter and the C concentrations in suspended sediment (Table 4)'. We agree it is difficult to discriminate strictly that suspended sediment is of mineral origin in the agricultural catchments, but rather a mixture of organo-mineral complexes as already mentioned in L. 348-352. 'These results suggest that the sediment from the forest catchment is comprised of organic material with a high C:N:P ratio either originating from the forest floor or falling directly into the river. In contrast, sediments from the agricultural catchments are a mix of mineral and organic matter and fertilizer additions do not balance with the loss of carbon, nitrogen and phosphorus from the system.'

LL.395-400: nice to have compiled all the data shown in Table 6; of course, it is really valuable to compare your results with those found in similar/tropical environments. Just a random question: is it meaningful to compare these results with those found in Spain, for instance? Are these environments /land management modes comparable?

Response: Thanks for this. We believe it is meaningful to make a global comparison of our results with those outside the tropics, especially when comparing our results of low-input systems with highly intensified agricultural systems from temperate regions.

L.424 : again, you refer explicitly to "surface erosion processes", but how can you convince the readers that subsurface erosion is negligible in these steep catchments?

Response: Please see earlier response. We included a paragraph in L. 379-385. 'In the smallholder agriculture catchment, the lowered concentrations of sediment TC, TN and TP can be explained by sediment originating from the subsurface where nutrient concentrations are lower (Russell et al. 2001; Gellis et al. 2009; Wanyama et al. 2018). This was demonstrated by Stenfert Kroese et al. (2020a), in a sediment fingerprinting study, that the subsoil sources are of increased importance in the smallholder agriculture compared to the natural forest and the tea-tree plantation catchment due to exposure of subsoil to erosion processes.'

We are reporting the work of others in this section and not commenting on our own catchments. To make this clearer we have modified the position of the citations. 'Similar soil nutrient losses have been observed in other densely populated tropical agricultural regions cultivated on steep hillslopes in Uganda, Tigray (Ethiopia), Kisii District (western Kenya) through nutrient losses, caused in particular by surface erosion and insufficient use of fertilizer application (Lederer et al., 2015, Girmay et al., 2009, Smaling et al., 1993).

---

## Author Comment (AC2) · 15 Jan 2021

We would like to thank Reviewer #2 for the feedback provided to our manuscript. We will take their comments into account when revising the manuscript.

Anonymous Referee #2 The paper by Stenfert Kroese et al. addresses the impact of land use on particulate carbon and nutrient export from tropical montane catchments in the South-West Mau in Kenya and shows that soil fertility is lost with the conversion from natural forest to cultivated land. This study fills in an important knowledge gap on particulate nutrient export of tropical ecosystems in East Africa. The manuscript is mostly well written and clearly structured.

[Figure]

I recommend it for publication in SOIL after some revisions.

1) My main concern is the way the data is presented. In Figures 4 and 5 and Table 4, concentrations and ratios are presented for each sampling year separately. This might be prone to misinterpretation, as different seasons were sampled for each year. In Table 5 it is stated that the data are from wet period (2018) and drier period (2019). I think it's necessary to be consistent throughout the manuscript, thus, the same labelling is necessary for Figures 4 and 5 and Table 4. Related to this: While in the results differences between years/seasons are acknowledged, in the discussion the whole seasonality is neglected. Especially the C:N, C:P, and N:P ratios seem to differ during the different seasons. Might this indicate different sediment sources? I think this needs to be addressed shortly in the discussion.

Response: Thanks for this. We amended the labelling of Figure 4 and 5 and Table 4. Each sampling period is presented as '2018 wet period' and '2019 drier period' and the headings of Figures 4 and 5 and Table 4 were also amended to 'based on 13 22 sampling days for the sampling campaign from May October 2018 and 14 18 sampling days for the period April June 2019' to match with Table 5 and to clearly indicate the sampling periods.

We added a section in the discussion to address the differing ratios during the years 'The significant lower C:N ratio in the natural forest and tea tree plantations in 2019 compared to 2018 might indicate a reduction in organic matter content of the sediment sources during the drier period in 2019 compared to the wetter sampling period in 2018. The significantly higher C:P and N:P ratios in 2019 in the smallholder agriculture catchment suggest that the source of phosphorus originates from higher mineralization rates of organic matter and unused fertilizer from bare agricultural surfaces in the drier period of 2019 compared to the drier sampling period in 2018.'

2) Furthermore, the methods section lacks some more details: How where the data stored? Did you use an external datalogger or did the sensor log internally? The
stealing of power supply and subsequent data loss is mentioned, but it is not clear how the setup was powered. How long did you let the sediments settle before air drying the aluminum trays?

Response: We included a paragraph in the method section: 'The data is stored automatically on a data logger (con::cube, s::can Messtechnik GmbH, Vienna, Austria) and downloaded on a weekly to bi weekly basis. The data is additionally automatically uploaded to an online database, except for the site at the natural forest where there is no cellular network. The equipment is powered by solar panels and two batteries.' It is also mentioned that 'a more detailed description of sampling sites and instrumentation can be found in Jacobs et al. (2018)' (L. 149-152).

The sediment samples were allowed to settle up to 5 days before air drying. We amended the sentence accordingly: 'Sediment in suspension from all three sampling methods were allowed to stand for up to 5 days, then the supernatant was carefully removed, the remaining sediment water mixture was then placed in aluminium trays and air dried' (L. 198-199).

Specific comments: Introduction

3) p. 2 L52: What is the impact of an increased turbidity in streams?

Response: We included 'by increasing turbidity, which prevents light reaching aquatic plants'.

4) p. 3 L65ff: The presented results are from which ecosystems? All tropical?

Response: The results are from tropical and temperate regions. We included 'other studies in temperate and tropical regions' in the section to be more precise (L. 60-69).

5) p. 3 L69: "This is an important knowledge gap"

Response: We corrected to '. . . knowledge gap' (L. 72).

Methods

6) p. 9 L176: Integrating missing discharge data linearly does not seem right, if rainfall data is available discharge can be correlated to rainfall?

Response: We didn't integrate missing discharge data with the linear interpolation, only the sediment data was interpolated (L. 182).

7) p. 9 L179/180: Instead of calling it the "drier period of the start of the long rainy season" maybe use "onset of the rainy season"

Response: We corrected accordingly: 'the drier period of the onset of the long rainy season' (L. 185).

8) p. 11 L212: organic matter content

Response: corrected

9) p. 11 L219: I'm wondering how representative the yearly yields of sediment-associated TC, TN and TP are if you use only data from 3 sampling periods over two years. Or did you relate the C and nutrient yields to the turbidity data? If so, it is not clearly described.

Response: We did not relate TC, TN and TP concentrations to the long-term turbidity dataset. We acknowledge that the annual yields of sediment-associated TC, TN and TP are rather estimates due to the short sampling period. We included a paragraph to highlight the uncertainty of TC and nutrient yields; however, we believe it is important to keep the yield estimates in order to compare the different fluxes within the three catchments. 'Our sediment-associated TC, TN and TP yield estimations are uncertain. This is due to the sampling of the drier start of the long rainy season in 2019 and the short sampling period in both years. This might have resulted in missed sampling of storm events. Increasing the sampling frequency would improve our understanding of the particulate TC, TN and TP fluxes.'

10) p. 11 Data analysis: Which programs did you use for data analysis?

Response: We used R studio for data analysis. This was included in the manuscript as 'Data analysis was conducted with R software (R Development Core Team, 2017).'

Results

11) p. 12 L235: State somewhere in the text that the values between the brackets are the 95% CI.

Response: Included 95%-confidence interval in brackets.

12) p. 12 L236: Define the catchment runoff coefficient and how you calculated it in the text

Response: We included a sentence in section 2.2 'To relate the amount of runoff to the amount of precipitation received, the catchment runoff coefficient was calculated as defined as specific discharge as proportion of annual rainfall'.

13) Figure 3: This is a really nice figure, however, it is not discussed at all in the manuscript. For example, I'm wondering why SSY are a lot higher during the 2019 season compared to the 2018 season in the NF and TTP catchment, although the discharge seems lower. I see that this has been more the focus of recent work by the authors and is not the aim of the present manuscript, however, in my opinion if the data is presented like this in the manuscript it should be discussed accordingly.

Response: We inserted a paragraph in the discussion on the annual suspended sediment yields but also referred to a more detailed discussion to Stenfert Kroese et al. (2020b). 'The wetter 2019 might have resulted in higher annual suspended sediment yields for the natural forest and tea tree plantation catchments. In contrast, the smallholder agriculture catchment experienced higher suspended sediment yields during a drier 2019 compared to the previous year. The late onset of the rainy season resulted in a late start of the cropping season. As discussed in Stenfert Kroese et al. (2020b) this might have left bare agricultural land prone to erosion during stronger, but shorter rainfall events.'
14) p. 15 L286-288: are the reported OM contents mean values over both years? Maybe add these values to a table?

Response: The reported OM contents are mean values of a few sediment samples from both years. We added the values to Table 1.

15) p. 17 L324-326: add C and N to the units to avoid confusion: kg C day-1 and kg N day-1. Also for TP in line 328.

Response: Thanks for this, we amended the text accordingly.

16) Section 3.4: Does it make sense to calculate mean annual yields of the sampling periods? How representative are these values? I think this needs to be addressed in the discussion.

Response: We acknowledge that the anual yields of sediment-associated TC, TN and TP are rather estimates due to the short sampling period. We included a paragraph to highlight the uncertainty of TC and nutrient yields; however, we believe it is important to keep the yield estimates in order to compare the different fluxes within the three catchments. 'Our sediment-associated TC, TN and TP yield estimations are uncertain. This is due to the sampling of the drier start of the long rainy season in 2019 and the short sampling period in both years. This might have resulted in missed sampling of storm events. Increasing the sampling frequency would improve our understanding of the particulate TC, TN and TP fluxes.'

Discussion

17) p. 19 L359: Do you know how long ago the conversion from forest to agricultural land occurred in your study sites?

Response: The conversion of the forest to tea-tree plantations started in the beginning of the 20th century, however the conversion to smallholder agriculture was during the last four to five decades. We amended the following sentence: 'Similarly, a decline in soil organic carbon and nutrients was observed following conversion to agricultural

cultivation in the same catchments of the Mau Forest Complex which was converted during the last four to five decades (Arias-Navarro et al., 2017; Owuor et al., 2018; Wanyama et al., 2018)'.

18) p. 19 L362: For an easier reading: put the values in the bracket for TC and TN concentrations behind "natural forest catchment". Also add C and N to the units.

Response: We amended this in the revised manuscript.

19) p. 19 L375: Figure 6 shows no strong correlation between TC and TP for SHA. Be precise that only TTP shows a significant relationship.

Response: We amended the sentence to 'A significant relationship was observed between TC and TP for the tea-tree plantation catchment, ...'.

[Figure]

**Fig. 1.** Figure 4

[Figure]

**Fig. 2.** Figure 5

---

## Author Response (AR2)

Response to the Author:

I think the authors did a good job in revising the manuscript and I have no bigger concerns for going ahead with the publication process.
Response: Thanks, we made the following changes.

One thing that I think still needs additional attention is the newly added text on differences in observations potentially stemming from the timing of the sampling (two different seasons in 2018 vs 2019 were sampled). I think this is an important caveat that has to be put more prominently also in the conclusion and the abstract which sum-up the study to the reader. But I see the changes in this regard only in the discussion. Please amend or elaborate why you think otherwise.
Response: We included the differences in the abstract and the conclusions as following: 'during a wet sampling period in 2018 and a drier sampling period in 2019' (L. 24 – 25) and 'The temporal differences in observations suggest to be stemming from the timing of a different sampling period in 2018 compared to 2019.' (L. 478-479).

Furthermore, the wording of the new text to describe 2018 v 2019 wetness differences is a bit confusing: for example, page 22: You start l.387 with "the wetter 2019" but in l.388 it is written "a drier 2019". I realize that these are different catchments, but I would advice to explain the discrepancies in the local weather conditions due to sampling and monitoring at different points in time already in the methods and then establish a clear wording how you call the seasonal differences. For example, in tables 2018 is described as the "wet period" but 2019 as the "drier period" instead of "dry period" or something similar. My advice would be to go through the manuscript and try to use clear and distinct wording whenever speaking of a dryer vs. wetter year, or a strong or weak dry/wet season.

Response: Overall, 2019 is wetter compared to 2018 in the natural forest and the tea-tree plantations. The smallholder agriculture is wetter in 2019 than 2018. However, the sampling period is wetter in 2018 and drier in 2019 for all three catchments.

We included one sentence about the sampling period in the methods: 'Suspended sediment was sampled during both long rainy seasons in 2018 and 2019 (May-September 2018 and April-May 2019). However, the sampling period in 2019 coincided with the onset of the long rainy season in 2019 and was drier compared to 2018. Therefore, the first sampling campaign in 2018 is described as '2018 wet period' and the second sampling campaign in 2019 as '2019 drier period' (L .188-191).

We further amended the sentence to 'The overall wetter 2019 with higher annual rainfall…' (L. 376) highlighting that we refer to the entire year and not the sampling campaign.

Minor comment for l. 240. Just say "Data analysis was conducted in R (or using R) ".
Response: Thanks, we amended accordingly.